# Promising Oxygen- and Nitrogen-Rich Azidonitramino Ether Plasticizers for Energetic Materials

**DOI:** 10.3390/molecules27227749

**Published:** 2022-11-10

**Authors:** Dmitry B. Vinogradov, Pavel V. Bulatov, Evgeny Yu. Petrov, Pavel S. Gribov, Natalia N. Kondakova, Natalia N. Il’icheva, Evgenia R. Stepanova, Anatoly P. Denisyuk, Vladimir A. Sizov, Valery P. Sinditskii, Aleksei B. Sheremetev

**Affiliations:** 1Zelinsky Institute of Organic Chemistry, Russian Academy of Sciences, 47 Leninsky Prosp., 119991 Moscow, Russia; 2Mendeleev University of Chemical Technology, 9 Miusskaya pl., 125047 Moscow, Russia

**Keywords:** azide, nitramine, dialkyl ether, energetic plasticizer, synthesis, impact sensitivity, phase transition, volatility, thermal decomposition, combustion, burning rate

## Abstract

A simple, mild and general method has been developed for the preparation of alkyl nitramines bearing a halogenoalkoxylic moiety. From these reactive halogen intermediates, a few azidoalkoxyl alkyl nitramines have been produced as energetic plasticizers. This simple protocol allows azidonitramino ether plasticizers to be obtained from available precursors in high yields, as it is safe and viable for large-scale operations. The resulting products have been fully characterized by spectral methods, and their impact sensitivity, thermal transformations and burning properties were determined, thus allowing complete comparison to the analogues including other combinations of structural units. Such characterization of these new plasticizers illustrates the extent to which the nature and position of the functional units can be used to tune the above properties of these nitramines. All azidonitramino ethers are liquid with excellent energetic performance and are promising candidates for new environmentally friendly energetic materials.

## 1. Introduction

The unifying basis of most multicomponent energetic materials is a binder consisting of a polymer and a plasticizer. Previously, both an inert polymer and a plasticizer were disposed of for these purposes [1]. In recent decades, the design and synthesis of diverse energetic polymers [2,3,4,5,6,7] and plasticizers [8,9,10] have received considerable attention. The use of an energetic binder increases energy output further than traditional binders, which obviously improve the performance of energetic materials. Strategically, this is usually realized by the incorporation of various explosophoric units [11] in polymer chains and/or in plasticizer backbones. Up to 80% of the binder can be occupied by a plasticizer, which thus can have a more significant effect on the performance of the energetic material than the polymer. For a specific energetic material, creating an optimal binder that, in combination with other components, would be suitable for processing both an uncured mass and a cured charge with the desired physical and mechanical properties is quite a difficult task.

Different plasticizers [12] give unequal plasticization effects due to the different strength of the plasticizer–polymer and plasticizer–plasticizer interactions. In order for the plasticizer to be effective and useful when introduced into a polymer, it must incorporate two types of structural units—polar and non-polar. The balance between the polar and non-polar parts of the molecules is very important for regulating the effect of the plasticizer on a particular polymer. An imbalance leads to inefficiency or even incompatibility. Only with a wide range of polymers and plasticizers in hand is there a chance to create a suitable binder.

To maximize its potential, a conventional plasticizer should [12]: (i) be thermodynamically compatible with the polymer; (ii) not be lost during use either by volatilization or by extraction from the binder matrix; (iii) have no odor; (iv) be chemically inert; (v) have a decomposition temperature that should not be lower than the polymer processing temperature; (vi) have affordable cost, and so on. A plasticizer reduces the glass transition temperature (T_g_) and the softening point of the polymer, increases its elasticity and strength and thus facilitates processing.

To date, numerous energetic plasticizers containing such explosophoric groups as nitro, nitrato, nitramino, or azido at various molecular backbones have been designed and synthesized [8,9,10]. On the one hand, the incorporation of the above explosophores tends to improve the density, enthalpy of formation and/or oxygen balance, and thereby results in an excellent energy performance. On the other hand, excessive accumulation of explosophoric groups on a compact backbone leads to increased sensitivity to thermal and mechanical stimulation and increases the polarity and reduces the flexibility of the target molecule, complicating its use for polymer plasticization. While the structural variety of energetic plasticizers has increased dramatically, there still remains a need for a broader investigation of diverse backbones and functional group combinations. New functionalized compounds with a novel set of properties are required for the development of advanced materials. Obviously, the more various combinations are synthesized and the more data on the properties of the prepared compounds are collected, the more important reference material for screening and design will be available to researchers.

In our continuing effort to seek new improved components for modern energetic materials, here we present our attempts at designing and synthesizing a series of energetic plasticizers that combine azido and nitramino groups on a dialkyl ether backbone.

While azidoalkyl nitramines are widely used as plasticizers [8,9,10,13], some shown in Figure 1, the effect of the incorporation of the ether bridge into these molecules has been studied very sparsely, and only meager physical and energetic properties are available [13,14,15,16,17]. On the other hand, due to the inherent stability and flexibility of the ether bridge, it is widely used for the development of generally applicable plasticizers suitable for operation in a wide temperature range. Moreover, the ether bridge, which has non-bonded electron pairs available for inter- and intramolecular interactions, provides a sufficiently low migration of the plasticizer from the binding matrix.

From a practical point of view, the synthetic protocol to the target compounds should be simple and concise, safe and scalable to provide rapid access to large quantities of material. It is important that the starting reagents are either inexpensive, commercially available or can be obtained using simple reliable processes.

Following the above guiding principles, we report the facile and effective synthesis of new azidonitraminoethers. The same protocol provided improved the preparation of analogues, which have previously been obtained but were insufficiently characterized. In addition, we report the reliable preparation of several azidoalkyl nitramines and nitrates as comparison compounds. These azides are a good set which enable investigation of the structure–property relationship in energetic plasticizers. In the context of our interest in various combinations of structural elements in energetic molecules, we also report extensive studies of the physical and special properties of all compounds. These azidonitraminoethers are valuable components for use in energetic materials and may also prove useful as intermediates in the synthesis of other materials.

## 2. Results and Discussion

### 2.1. Synthesis

In the course of our work on functionalized nitramines [15,17,18,19,20,21,22], we focused on available chloromethylnitramines **1** and envisioned nucleophilic substitution of chlorine by haloalcohols, which would give nitramino ethers **2** the halogen atom in which can be replaced by an azide group (Figure 1). This expectation was based on the possibility, reported by Frankel et al. [13], of replacing the chlorine atom in chloromethylnitramines with sterically unencumbered chloroethanol, which occurred during prolonged refluxing (up to 120 h) in dichloroethane (DCE) and gave a crude mixture contained ca. 60% of the target product requiring chromatographic purification.

A well-known trend is that an increase in the molecular weight and branching of the plasticizer molecule favors more acceptable properties. We commenced our study with N-(chloromethyl)-N-methylnitramine **1a** [22] and more sterically hindered alcohol, 1,3-dichloropropan-2-ol. The desired ether **2a** was indeed formed during refluxing in DCE. We found that the yield of ether **2a** is strongly dependent on the reaction time, as determined by monitoring the reaction mixture by ^1^H NMR. The maximum accumulation of product **2a** in the reaction mixture was fixed after 12 h of heating; with further refluxing, the yield decreased, and by-products were formed.

The pure ether **2a** was easily isolated in good yield after a simple washing of the crude product with an aqueous 10% NaHCO_3_ solution at 70–75 °C. With a slight excess (up to 10%) of precursor **1a**, the yield of product **2a** is 76%.

A related reaction occurred using 2,3-dibromopropan-1-ol and 3-bromo-2,2-bis(bromomethyl)propan-1-ol, resulting in the formation of ethers **2b** and **2c**, respectively. Using the similar protocol, **2b** and **2c** were obtained in 84% and 71% yields, respectively.

It is noteworthy that a review of the literature using SciFinder did not find an ether bearing 2,3-dibromopropyl-1 moiety. To the best of our knowledge, there are no reports describing the preparation of 3-bromo-2,2-bis(bromomethyl)propan-1-ol ethers in the absence of a base; KOH or NaH are typical bases, and the reaction is carried out in DMSO or DMF [23,24,25].

A similar technique to that used for ethers **2a**–**c** also gave excellent results for the synthesis of diether **2d**. Although the yield of product **2d** is close to what was reported earlier [13], the reaction time is shorter (12 h versus 24 h) and the purification is simpler.

Azidation is an extensively studied fundamental reaction of great industrial importance, the products of which are key precursors in organic synthesis or target energetic compounds [26,27,28]. Depending on the reactivity of the starting substrates, a host of synthetic protocols have been developed. When alkyl azides are prepared from alkyl halides, the nucleophilic substitution of a halogen or equivalent leaving group is typically performed at heating with sodium azide in DMSO or DMF. The work-up of such reaction mixtures usually involves dilution with water, extraction into an organic solvent (for example, CH_2_Cl_2_/CHCl_3_/EtOAc/Et_2_O) and subsequent chromatography or distillation giving the target azide. However, the protocol suffers from problems in product recovery, accompanied by a loss of yields. From the point of view of synthesis, water is a very attractive solvent for the preparation of energetic compounds, especially such dangerous ones as azides.

The replacement of organic solvents with an aqueous medium is of great interest for the chemical industry [29,30,31,32] and is particularly promising for reducing risks in the synthesis of energetic compounds. However, difficulties may arise due to the insolubility of the starting reactants, and phase transfer catalysis (PTC) [33] is very popular to eliminate this drawback.

With this in mind, we have studied the possibility of obtaining alkylazides in an aqueous medium. Preliminary work in our laboratory has shown that the azidation reaction of chloralkyl nitramines in water is compatible with PTC [17].

We initiated our studies by evaluating the reaction of dichloride **2a** with NaN_3_ under a variety of conditions (Figure 1). The phase transfer catalysts studied included tetraalkylammonium chlorides and bromides. Overall, we found that tetraalkylammonium bromides are more efficient than the corresponding chlorides, providing higher yields and shorter reaction times. We established that a small excess of NaN_3_ (1.1 equiv per halogen) and the addition of 0.1 equiv of tetraalkylammonium bromide was required to achieve complete conversion of dichloride **2a**. Accordingly, the reaction of **2a** with NaN_3_ in the presence of tetrabutylammonium bromide in water under reflux for 12 h providedthe desired N-((1,3-diazidoprop-2-oxy)methyl)-N-methylnitramide **3a** in 87% isolated yield. When this reaction was carried out using more expensive tetraalkylammonium bromides, product **3a** was formed with comparable yields over a similar time. Significantly, diazide **3a** could be synthesized in multigram quantities, with only extraction and simple washing with water necessary to obtain the desired product in pure form.

Pleasingly, the similar azidation of other, even more branched haloalkylnitramines was also successful, allowing for the synthesis of compounds **3b**–**d** with good yields (Figure 1). However, some reactions take longer to complete.

In addition to ether-based azidonitramines, we synthesized energy plasticizers without an ether bridge as comparison compounds. So, a recent patent of Chinese researchers showed that N,N-bis(2-azidoethyl)nitramide **6** can be built up from N,N-bis(2-chloroethyl)nitramide by azidation in water for 6 h in 80% yield under PTC [34]. By replacing chlorine with bromine in the precursor, synthesizing bromo analog **5** (Figure 2), we showed that when it was treated with NaN_3_ under the conditions we found, compound **6** can be obtained in 95% yield in a shorter time (3 h).

Available nitramino oxetane **7** [35] was converted to branched diazide **10** in three steps as shown in Figure 2. The ring-opening of the oxetane **7** was achieved by acid catalysis to form the diol **8** in 85% yield. The product **8** was then treated with p-toluenesulfonyl chloride (TsCl) in the presence of a base to afford tosylate **9** in 92% yield. Finally, the tosyl groups of compound **9** were substituted with an azide nucleophile to furnish target **10** in 89% yield.

1,3-Diazidopropan-2-ol **12** [36,37,38] had previously been prepared by various protocols and used as a starting material in the synthesis of nitrate **13** [39,40]. Despite the fact that 1,3-dichloropropan-2-ol **11** is soluble in water, azidation in the presence of tetrabutylammonium bromide proceeds faster than in its absence, and product **12** is formed in quantitative yields (Figure 2). Following known procedures [39,40], the nitration of **12** was carried out with HNO_3_/Ac_2_O. However, the literature procedure was modified by using a lower temperature for the nitration. We were pleased to find that with HNO_3_/Ac_2_O, this reaction provided a 92% yield of the desired product **13** after 20 min at −10 °C. For comparison, 72% yield was achieved at 0–5 °C in 3 h [40], and 85.6% at 10–15 °C/45 min [39].

The developed methods are scalable, and all azides of this study could be synthesized in multi-gram quantities.

### 2.2. Sensitivity Measurements

An important property of an energetic material is its sensitivity to mechanical stimuli [41]. In order to determine the hazards associated with the compounds of this study, the samples were analyzed for impact and compared with those of the commonly handled energetic plasticizer nitroglycerine (**NG**) (Table 1). Sensitivity is characterized as the percentage of explosions at impact (10 kg drop weight at a height of 25 cm, 25 tests); the higher the percentage of explosions, the more sensitive the sample. By our testing methods (K-44-II impact machine, set No.1 [42], which is similar to STANAG 4489 Test 3(a)(ii)), compound **13**, like **NG**, has a maximum impact sensitivity (IS) value of 100%. Azide **3d**, which has the lowest burning rate, has the lowest IS. Compounds **3b** and **10**, whose burning rates are slightly higher, also have a higher sensitivity. As expected, the more azide groups the compound includes, the higher the sensitivity. Thus, triazide **3c** is 16% more sensitive than diazide **3a**.

### 2.3. Relaxation and Phase Transitions

For azides of this study, the low-temperature analysis using differential scanning calorimetrics (DSC) (closed aluminum crucibles, the sample was cooled uncontrollably with liquid nitrogen, ramp rate = 10 °C min^−1^) were performed to determine the temperature for glass transition (T_g_), crystallization (T_c_) and melting point (T_m_). Typical step-by-step phase transitions from the solid state to the isotropic melt of compound **6** were observed in the temperature range from −110 to 30 °C as illustrated in Figure 2.

At −100 °C, azide 6 is an amorphous crystalline compound. When heated to ca. −88 °C, de-vitrification occurs, and changes in heat capacity are recorded on the DSC plot. Upon further heating, in the range from −48 to −41 °C, cold crystallization of the sample occurs, manifested by an exothermic peak, and then melting, which is evidenced by an endothermic peak (Figure 2). Azide **3d** also belong to amorphous crystalline compounds (see, Appendix A). Unlike azide **6**, which melts ca. 30 °C above the temperature of the end of crystallization, the melting of compound **3d** begins immediately after crystallization.

Azides **3a**, **3b**, **3c**, **10** and **13** are amorphous compounds. In these cases, only one phase transition associated with the relaxation process of glass transition was observed (see Appendix A). It is well known that some compounds are not prone to crystallization and remain in a supercooled state for a long time. This may be due to crystallization conditions, or to the ability of the compound to form polymorphs that give eutectic mixtures. For example, during spontaneous crystallization of **NG**, a stable phase with a melting point of +13 °C is formed [43]. However, it remains in an amorphous state and neither the peak of crystallization nor the peak of melting is recorded on the DSC when heated using a 10 °C min^−1^ ramp rate (see, Appendix A). If the sample is heated at a low rate of 1.25 °C min^−1^, not only the transition from a glassy to an amorphous state, but also cold crystallization and a melting point (1 °C) are observed (see Appendix A). This melting point characterizes crystals of labile **NG** form.

The temperatures of glass transition, crystallization and melting, as well as the heat effects of phase transitions for azides of this study, are listed in Table 2.

It should be noted that the widely used benchmark plasticizer, nitroglycerin (**NG**), has a glass transition temperature of −70 °C. Only azide **3d** has a glass transition temperature 9 °C higher than that of **NG**. The glass transition temperature of the other azides of this study is lower than that of **NG**. The absence of crystallization and low glass transition temperatures allow us to consider azides **3a**, **3b**, **3c**, **10** and **13** as possible plasticizers of frost-resistant energetic materials.

### 2.4. Volatility

The volatility of the plasticizer determines the consistency of the composition and properties of materials containing these components, as well as the conditions for safe work with them. However, most liquid energetic plasticizers have rather high volatility. In particular, this disadvantage is also inherent in the widely used **NG**. Energetic plasticizers less volatile than **NG** are of great interest.

Here, the volatility of organic azides **3a**, **3b**, **3c**, **6**, **10** and, for comparison, **NG** was determined by thermogravimetric analysis (TGA). The assessment of the volatility of compounds is based on the measurement of the saturated vapor pressure or the rate of weight loss per unit of the evaporation surface at a given temperature [45,46,47]. The relationship of the volatility parameters is described by the Langmuir Equation (1):(1)$$G=\frac{dm}{dSdt}=kP\sqrt{\frac{M}{2\pi RT}}$$
where *G* = $\frac{dm}{dSdt}$ is the evaporation rate (loss of weight *m* from a unit of surface *S* to a unit of time *t*), *k* is the evaporation coefficient, *P* is the saturated vapor pressure, *M* is the molecular weight, *R* is the universal gas constant and *T* is the temperature.

When a compound evaporates in a vacuum, it is assumed that *k* = 1. During evaporation in a gas atmosphere, this coefficient depends on the conditions of mass transfer of molecules of the evaporating compound in the gas atmosphere (the rate of diffusion of molecules in the gas layer near the evaporation surface, the speed and flow regime of the gas flow, the composition and pressure of the gas, etc.). A comparative method for determining the vapor pressure was previously proposed [48]. This method combines the evaporation data obtained by the TGA method and the available data on the vapor pressure of calibration compounds obtained by an independent method. This makes it possible to exclude the evaporation coefficient *k* from the calculations. To estimate the volatility, the relative volatility coefficient F*_rel_* can be used, which characterizes the ratio of the vapor pressure of a given compound to the vapor pressure of the compound accepted as standard according to Equation (2) [48]:(2)$$F_{rel}=\frac{P_{l}}{P_{st}}=\frac{G_{l}}{G_{st}}{\left(\frac{M_{l}}{M_{st}}\right)}^{0.5}$$
where the indices *l* and *st* refer to the compound under study and the compound taken as standard, respectively. We used **NG** as the standard.

Thermograms characterizing the change in the mass of the azides of this study and NG during heating in the nitrogen-purged cell of the thermal analyzer scales are shown in Figure 3. Preliminary TGA and DSC data showed that there is no decomposition of these compounds in the temperature range from 25 to 80 °C. Thus, the decrease in the mass of the samples is associated only with evaporation.

The evaporation rates of samples were calculated by the time differentiation of thermogravimetric curves and their subsequent mathematical processing. The calculated evaporation rates of selected compounds and their relative volatility coefficients are summarized in Table 3. The table clearly demonstrates that the volatility of the studied azides is significantly lower than that of **NG** and diethylene glycol dinitrate (**DNDEG**).

In particular, the volatility of azides **3a** and **3c** is five times lower than that of **NG**. Not only the molecular weight, but also the structure of the molecule affects the volatility. This is clearly seen when comparing isomeric azides **3a** and **3b**; the volatility of branched azide **3a** is 3.5 times less than that of its linear isomer **3b**.

### 2.5. Thermal Analysis

The thermal stability for compounds of this study was determined by differential scanning calorimetric (DSC) and thermogravimetric analysis (TGA) measurements scanning at 10 °C min^−1^ (Table 4, Appendix A). Compounds **3d** and **6** are low-melting solids, while all other azides are liquids at room temperature. According to the TGA, despite the fact that the experiment was carried out in aluminum crucibles closed with a lid, all samples almost completely evaporated to the decomposition temperature, which is observed by DSC. In the crucible of DSC, compound **13** begins to evaporate intensively before decomposition; as a result, the peak of heat absorption during evaporation is superimposed on the peak of heat release during decomposition. Despite the overlapping peaks, it is obvious that compound **13** is the least heat-resistant. This is due to the presence of a nitrate ester group, which initiates the decomposition of the compound. The decomposition temperatures of the other azides of this study ranged from 241 to 257 °C (Table 4). Evaporation undoubtedly affects the amount of heat released, which varies from 152 to 840 kJ mol^−1^ for diazides and is 603 kJ mol^−1^ for triazide **3c**.

Since the thermal decomposition effect of the azide group in non-volatile compounds, for example, glycidyl azido oligomers, is 165–170 kJ mol^−1^ [49], higher heat effects for **3d** (840 kJ mol^−1^) and **3c** (603 kJ mol^−1^) indicate that the decomposition of both azide and nitramine groups takes place.

There is an extensive literature describing the thermal transformations of organic azides [50,51,52]. Alkyl azides are the most thermally stable in this class of compounds. It was previously suggested that an inductive effect has an important effect on the stability of alkylazides [51]. Indeed, the azides of this study, where the azido group is bonded to the electron-withdrawing nitramine group by an alkyl bridge (compounds **6** and **10**), are slightly less stable than their analogues (compounds **3d** and **3c**), whose bridge includes an ether bond which has negative inductive and positive mesomeric effects. The azido ester **3b** falls out of the last group, which is most likely due to the mutual influence of the azide groups, which in this compound are linked by the shortest aliphatic bridge.

An isothermal decomposition study of azides **3a**, **3c** and **6** was performed using a Bourdon glass compensation pressure gauge [53]. The ratio of the mass of a sample to the volume of the reaction vessel (m/V) was ~4 × 10^−3^ g cm^−3^. The dependence of gas released during the decomposition of the above compounds on time was determined (Appendix A).

The thermolysis of compounds **3a** (150–190 °C) and **6** (150–175 °C) is described by the first order reaction. The decomposition of compound **3c** (170–190 °C) proceeds with a slight acceleration, so its gas release data were processed by a first-order model with autocatalysis [53]. The final volume of gases released from **3a** is 390 cm^3^ g^−1^ (3.8 moles per mole of **3a**); 510 cm^3^ g^−1^ (4.55 moles per mole) was released from compound **6**, and 420 cm^3^ g^−1^ (5.6 moles per mole) was released from **3c**. The decomposition rate constants could be described by the equations k = 10^14.36^ exp(−161,540/RT), c^−1^ (I), k = 10^14.28^ exp(−162,040/RT), c^−1^ (II) and k = 10^15.07^ exp(−172,570/RT), c^−1^ (III) for azides **6**, **3a** and **3c**, respectively.

Previously published data for azide **6** obtained under non-isothermal conditions using the Ozawa method give a significantly lower activation energy of 115.3 kJ mol^−1^ [44]. The stability of azide **6** is comparable to that of the previously studied promising energetic material 4,4′-bis(azidomethyl)azofurazan [54].

The decomposition rate constants of azides **3a**, **3c** and **6** under isothermal conditions are in good agreement with the data at higher temperatures (non-isothermal conditions, DSC). Isothermal decomposition rates for **6**, **3a** and **3c** were determined and evaluated by the Kissinger equation [55], using the activation energies from the manometric experiments. Figure 4 also shows the literature data on the decomposition of two benchmark azides, glycidyl azido polymer (**GAP**) [56] and 1,3-diazido-2-nitro-2-azapropane (**DANP**) [52], illustrating the opposite limits of the inductive effect. In **GAP**, the azide groups are separated by an alkyl ether bridge of six atoms that do not transmit the inductive effect well, while in **DANP**, the azide group is bonded to the electron-withdrawing nitramine group by only one CH_2_ unit. It is clearly seen that the thermal stability of compound **3c** and **GAP** are close. The decomposition rate constants of azides **6** and **3a** are only 2–3 times greater than the rate constant of azide **3c**, but more than 30 times higher than that of the least stable **DANP**. A similar trend is observed for the activation energy, which decreases in the following order: **3c** (172.6 kJ mol^−1^) > **6** ≅ **3a** (161.5–162.0 kJ mol^−1^) > **DANP** (151.0 kJ mol^−1^ [52]).

Based on the decomposition kinetic parameters, it is possible to predict thermal safety, without which it is impossible to assess the prospects for practical application. Firstly, the degree of decomposition (*η*) of azides when stored at room temperature can be estimated using the equation: $\eta =(1-e^{-k\tau })$ 100, %, where *k* is the decomposition rate constant of a sample at 298 K, and *τ* is the storage time of 20 years (or 6.3 × 10^8^ s). Secondly, the self-accelerating decomposition temperature (SADT) under adiabatic conditions can be estimated. In accordance with the United Nations standard [57], the SADT is defined as the lowest ambient temperature at which a temperature raise in the bulk of a particular package exceeds 6 °C after a time period of 7 days. The degree of decomposition of the sample can be calculated based on its Arrhenius parameters. To calculate the temperature increase, the heat of reaction (Q) determined in the DSC experiments (Table 4) and the heat capacity of 1.67 J·g^−1^ K^−1^ were used. The results are shown in Table 5. The ignition points of azides (sample of 0.05 g) when heated from 100 °C using a 20 °C/min ramp rate are also displayed in Table 5.

As can be seen from Table 5, the decomposition rate constants of azides in this study at a temperature of 298 K are significantly lower than that of the benchmark plasticizer, nitroglycerin (**NG**), and are comparable to the stability of nitrocellulose (**NC**). At this temperature, only thousandths of a percent of these azides will decompose over 20 years of storage. The azides of this study, judging by the calculated self-accelerating decomposition temperature, can withstand higher storage temperatures, and therefore, no temperature restrictions need be imposed on their storage and use.

The values of the ignition points of azides presented in Table 5 give reason to believe that this parameter, along with the decomposition kinetics, is influenced by the magnitude of the thermal effect of their decomposition. In particular, azide **3c** is the most stable, but has a low decomposition heat, so its ignition point is close to those of other less stable, but more energetic azides. It is clearly seen that for all azides of this study, the ignition point is higher than that of **NC** by ~30–50 °C.

### 2.6. Combustion

It is well known that liquid alkyl azides [49,59,60], like other liquid energetic compounds [58], are characterized by a light transition of combustion from laminar to turbulent mode. Andreev reported [58] that turbulence can have a double effect: (i) increases the apparent burn rate due to an increase in the burning surface, and (ii) prevents combustion, leading to its attenuation, due to the destruction of the heated layer of the condensed phase. Violations of the liquid–vapor interface (turbulence of the burning surface) depend on the viscosity of the liquid. To eliminate this defect, the viscosity of these liquids can be increased by thickening them with a polymer, such as nitrocellulose (**NC**). Here, the burning rate for the azides of this study was determined on samples that were pre-thickened with 4% **NC** (colloxylin 12% N). The dissolution of nitrocellulose was carried out from 1 to 2 h at 50–60 °C into transparent acrylic tubes of 7 mm i.d.

Burning rate data of azides **3b**, **3d** and **10** were determined in a wide range of pressures (Figure 5a). Burning rates of these azides and **NC** (12% N) are close, but significantly lower than that of **NG**. The compound **3d**, which has the longest carbon chain among the azides of this study, burns slightly slower than **NC**. At a pressure of 2 MPa, burning rates for all azides of this group and **NC** are the same.

Within the limits of the experimental error, burning rates of compounds **3b** and **10** are the same, and due to a higher index in the law of combustion, (n = 0.98) exceed the burning rate of **NC** at high pressures. Burn parameters of liquid azides are given in Table 6.

For nitroxazide **13**, the laminar combustion mode could not be implemented. After ignition, this compound burns at a very high rate, significantly exceeding that of **NG**. As noted above, compound **13** is volatile and the least stable of the azides in this study.

Experimental data on the combustion of compounds **3a**, **3c** and **6** are presented in Figure 5b. These azides burn only at high pressures. For **3a**, only a transitional site with a very large index n ~ 2.7 was observed, whereas for **6** and **3c**, small sites of laminar combustion (with an index n = 1.08 and 1.0, respectively) were recorded, after which combustion switches to a turbulent mode and it is impossible to measure its rate at higher pressures. The ability to combustion only at high pressures was previously observed for GAP liquid oligomers [49,60]. This is due to the fact that at high pressures, turbulent combustion, which is inhibited due to destroying the preheated layer of the condensed phase, changes due to an increase in the density of the outgoing gaseous combustion products supporting the laminar regime. However, the increase in the rate with increasing pressure continues, and for compounds with a high index n, there comes a moment when the combustion breaks down again to a turbulent mode.

Thus, all azides of this study are capable of self-sustained combustion, which, as in the case of other liquid energetic materials, easily switches to turbulent mode. However, in solid mixed compositions based on these azides, such a transition will be difficult, and their combustion will proceed in laminar mode.

The volatility of azide plasticizers suggests that they, like **NG**, burn according to the mechanism of volatile compounds [58,61], that is, the rate-determining reaction is in the gas phase, the heat flux from which warms up the compound to the boiling point and evaporates it. In this case, the lower burning rates of azides compared to **NG** are most likely due to the lower temperature of the leading flame zone in which azides decompose. It was previously shown [49,60] that in the absence of an oxidizer, the azide group does not release all the energy stored in it, since endothermic combustion products including double and triple C-N bonds are produced. Since the nitramine group, as an oxygen carrier, is more stable than the azide group, in the combustion wave, it decomposes at higher temperatures and has no effect on the burning rate.

### 2.7. Plasticizing Ability

Compounds bearing azide and nitramine groups are usually compatible with **NC** and other energetic polymers (**GAP**, **BAMO**, etc.) [62,63]. For modern propellants operated in a wide temperature range, an important characteristic is the transition temperature of the binder, which is a mixture of polymer and plasticizer, from a highly elastic state to a glassy one. The crystallization of plasticizers leads to a limitation of the lower limit of the temperature of the use of a propellant [64,65].

An early study based on an interference diffusion method [66] showed that azide **6** has unlimited thermodynamic compatibility with a polyether urethane polymer [67]. A binder, based on the polyether urethane polymer plasticized with nitroester plasticizer 1,2,4-butanetriol trinitrate (**BTTN**), demonstrates excellent low-temperature deformation ability [68]. Here, the plasticizing ability for the azides of this study with respect to **NC** (12.2 N%) and polyester urethane polymer (**PU**) synthesized on the basis of polyethylene-butylene glycol adipinate and 2,4-toluenediisocyanate cured with 1,3-butanediol [69] was determined. The content of the plasticizer in the mixture with **NC** was equal to 50 wt%, whereas in mixtures with **PU** it was changed from 40 to 90 wt%. The mixing of polymers with a plasticizer, such as liquid azide or, for comparison, **NG**, was carried out by direct swelling. The glass transition temperature of the resulting binders was determined by DSC and analyzed with a temperature ramp of 10 °C min^−1^ (Figure 6).

The glass transition temperature of **NC**—azidoplasticizer (50/50) is higher than that for **NC**–**NG** mixture by 4–24 degrees, depending on the azide used. In the series of azides, the glass transition temperature of **NC** is most effectively reduced by compound **3a**. While many of the azides of this study have low intrinsic glass transition temperatures (Table 2), in a mixture with **NC**, only compounds **3a**, **3b** and **10** provide an acceptable resulting T_g_ (Table 7).

For the plasticization of **PU**, azides **3a**, **3b**, **6** and **10** were used in comparison with **NG**. Figure 7 shows the dependence of the glass transition temperature of **PU**-based binders on the percentage of plasticizer.

The glass transition temperature of **PU** decreases sharply with an increase in the content of azide **10** to 60%. However, with a further increase in the content of azide **10**, from 60 to 70%, the T_g_ of the **PU**/azide **10** composition decreases by only one degree. With a percentage of plasticizer equal to 50 and 60%, the **PU**/azide **10** composition has a glass transition temperature 10 degrees lower than **PU** plasticized with **NG**. Azide **6** reduces the glass transition temperature of **PU** most effectively; at 70% of compound **6** in a mixture with **PU**, T_g_ of such a binder is reduced to −80 °C.

In terms of the effectiveness of reducing the glass transition temperature of the binder, the established order of efficiency is **6** > **3a** ≅ **3b** > **10**. All azides reduce the glass transition temperature of **PU** by more than **NG**. Thus, to create **PU**-based binders, nitramines bearing azide groups can be used as plasticizers. The percentage of such plasticizers can reach 70%, at which point the binder is in a highly elastic state over a wide temperature range.

### 2.8. Physical and Energy Properties

Although the data in Table 1, Table 2, Table 3, Table 4, Table 5, Table 6 and Table 7 demonstrate a number of properties of the compounds of this study, which make it possible to assess their advantages in comparison with benchmark energetic plasticizers, some more characteristics should be discussed.

The properties of energetic compounds depend on the oxygen and nitrogen content in their composition; the disadvantage of one can be compensated by another. Table 8 compares the oxygen coefficient and nitrogen content of the azidonitramino ethers with **NG** and azidonitramino alkanes, **EtAENA**, **BuAENA** and **DANPE** (**6**). Among these compounds, **NG** is the most oxygen-rich, but is nitrogen-poor. The compounds of this study have a more preferable content of both oxygen and nitrogen than the benchmark azidonitramino alkanes. While the densities of non-energetic plasticizers are typically below 1, the densities of azidonitramino plasticizers range from 1.262 to 1.366 g cm^−3^. 

An important characteristic for the components of composite and double-base propellants is the enthalpy of formation [60,70,71].

Unlike **NG**, which has a negative enthalpy of formation, all of the new compounds exhibit a positive enthalpy of formation ranging between 1.01 and 2.31 kJ g^−1^. Adiabatic flame temperature (T_f_) for azidonitramino ethers is lower than that of **NG**, which is attractive for wide application.

## 3. Materials and Methods

IR spectra were recorded on a BrukerALPHA instrument in KBr pellets. The ^1^H, ^13^C, and ^14^N spectra were recorded on a Bruker AM-300 instrument (300.13, 75.47 and 21.69 MHz, respectively) at 299 K. The chemical shifts of ^1^H and ^13^C nuclei were reported relative to TMS, for ^14^N, relative to MeNO_2_, high-filed chemical shifts are given with a minus sign. Elemental analysis was performed on a PerkinElmer 2400 Series II instrument. Analytical TLC was performed using commercially pre-coated silica gel plates (Kieselgel 60 F_254_), and visualization was affected with short-wavelength UV-light.

Most of the reagents and starting materials were purchased from commercial sources and used without additional purification. The starting chloromethylnitramines [22] were synthesized by using previously reported procedures.

Thermal stability, relaxation and phase transitions were studied by differential scanning calorimetry (DSC) using a Mettler Toledo DSC 822e module. Approximately 2 mg of the compounds was weighed into a placed in a 40 µL aluminum crucible, sealed under air with the appropriate sample press, and then pierced with a needle to leave two holes of approximately 1 mm diameter. The decomposition of a sample was carried out in a nitrogen atmosphere at a purge rate of 50 μL min^−1^. The temperature of the onset of intense decomposition (T_onset_) was taken as the temperature determining thermal stability. To study relaxation and phase transitions, the samples were uncontrollably cooled to −130 °C and then heated at a rate of 10 °C min^−1^. The temperature of the midpoint of the relaxation transition was taken as the glass transition temperature (T_g_). The glass transition process is accompanied by a change in the heat capacity of the sample ΔCp, which was also measured. The melting point (T_m_) of the individual compounds was determined as the temperature of the melting effect start point. The volatility of organic azides was determined on a TGA/SDTA 850e module. Liquid compounds weighing from 160 to 190 mg were placed in a cylindrical aluminum cup with an inner diameter of 9.0 mm and a height of 2 mm (Figure 8a). The accuracy of measuring the mass of the sample is 0.01∙mg. Weight loss measurements were made over a temperature range of 25 to 80 °C at a heating rate of 1 °C min^−1^. The samples were subjected to thermostating in the measuring cell at a temperature of 25 °C for 30 min before the start of measurements.

The burning rate was determined in a constant pressure device (Crawford bomb) with a volume of 2 L in a nitrogen atmosphere. Liquid compounds were mixed with 4% nitrocellulose (colloxylin 12% N). The dissolution of nitrocellulose was carried out from 1 to 2 h at 50–60 °C into transparent acrylic tubes of 7 mm i.d. (Figure 8b) The combustion process of the sample was recorded using a pressure strain gauge, which transmitted the signal to a digital oscilloscope. The start and end times of combustion were determined from oscillograms. The burning rate was calculated by dividing the sample height by the burning time and was related to the mean integral pressure during the experiment. The error in determining the burning rate does not exceed 3%.

The impact sensitivity of the studied azides was measured with a K-44-II impact machine set No.1, [42] with a 10 kg drop weight and a height of 25 cm. The frequency of explosions was determined by the number of explosions from 25 tests performed.

**Caution!** Although we have encountered no difficulties during preparation and handling of these compounds, they are potentially explosive energetic materials. Manipulations must be carried out by using appropriate standard safety precautions.

**2-Nitraza-4-oxa-5-(chloromethyl)-6-chlorohexane** (**2a)** (**General Procedure**). To a solution of 1-chloro-2-nitrazapropane (**1a**, 10 g, 80.3 mmol) in DCE (80 mL) was added 1,3-dichloropropanol-2 (8.6 g, 66.67 mmol). The resulting solution was stirred under reflux for 12 h, and the solvent was removed under vacuum to an yellow oil (~16.6 g). The crude product and 5% NaHCO_3_ (~150 mL) were intensively stirred at 75 °C for 1 h. The mixture was extracted with benzene (2 × 100 mL,), and the combined organic layers were washed with water (50 mL) and dried (Mg_2_SO_4_). The benzene from the combined extract was evaporated under reduced pressure to give a light yellow oil **2a** in 76% yield (11 g, NMR purity of ≥95%). n_D_^22^ = 1.4973. IR, ν, cm^−1^: 1079 (C-O-C), 1300 (NNO_2_ sym.), 1532 (NNO_2_ asym.); ^1^H NMR, δ (MHz, DMSO-*d*_6_): 3.40 (s, 3H, CH_3_NNO_2_), 3.76 (qd, *J* = 11.7, 5.1, 4H, 2 × CH_2_Cl), 4.08 (m, 1H, OCH(CH_2_Cl)_2_), 5.36 (s, 2H, O_2_NNCH_2_O) (Appendix A); ^13^C NMR δ (DMSO-*d*_6_): 37.8 (CH_3_NNO_2_), 44.3 (2 × CH_2_Cl), 77.9 (OCH), 79.4 (NCH_2_O). Found %: C 27.75; H 4.58; N 12.79. C_5_H_10_Cl_2_N_2_O_3_. Calculated %: C 27.67; H 4.64; N 12.91.

**2-Nitraza-4-oxa-6,7-dibromheptane (2b).** Following the general procedure, 1-chloro-2-nitrazapropane **1a** and 2,3-dibromopropanol-1 gave the desired product in 84% yield: n_D_^22^ =1.5392; IR, ν, cm^−1^: 1083 (C-O-C), 1298 (NNO_2_ sym.), 1531 (NNO_2_ asym.); ^1^H NMR, δ (DMSO-*d*_6_): 3.39 (s, 3H, CH_3_NNO_2_), 3.92 (m, 4H, CH_2_O, CH_2_Br), 4.56 (dt, *J* = 10.4, 5.1, 1H, CHBr), 5.27 (s, 2H, O_2_NNCH_2_O) (Appendix A); ^13^C NMR δ (DMSO-*d*_6_): 34.9 (CH_2_Br), 38.4 (CH_3_NNO_2_), 50.8 (CHBr), 71.1 (OCH_2_), 80.6 (NCH_2_O). Found %: C 19.90; H 3.17; N 9.11. C_5_H_10_Br_2_N_2_O_3_. Calculated %: C 19.63; H 3.29; N 9.16.

**2-Nitraza-4-oxa-6,6-bis(bromomethyl)-7-bromheptane (2c)**. Following the same procedure, 1-chloro-2-nitrazapropane **1a** and 2,2-bis(bromomethyl)-3-bromopropanol-1 gave the desired product in 71.4% yield: mp 73–74 °C (Et_2_O); IR, ν, cm^−1^: 1096 (C-O-C), 1296 (NNO_2_ sym.), 1526 (NNO_2_ asym.). ^1^H NMR, δ (DMSO-*d*_6_): 3.40 (s, 3H, CH_3_NNO_2_), 3.51 (s, 6H, 3 × CH_2_Br), 3.56 (s, 2H, OCH_2_C), 5.24 (s, 2H, O_2_NNCH_2_O). ^13^C NMR δ (DMSO-*d*_6_): 34.7 (3 × CH_2_Br), 37.9 (CH_3_NNO_2_), 43.2 (^t^C), 67.5 (OCH_2_), 80.1 (NCH_2_O). Found %: C 20.48; H 3.01; N 6.77. C_7_H_13_Br_3_N_2_O_3_. Calculated %: C 20.36; H 3.17; N 6.78.

**1,12-Dichloro-3,10-dioxa-5,8-dinitrazadodecane (2d).** Following the same procedure, 1,6-dichloro-2,5-dinitrazahexane **1b** and 2-chloroethanol-1 gave the desired product in 71.5% yield: mp 42–43 °C (Et_2_O). IR, ν, cm^−1^: 1091 (C-O-C), 1275 (NNO_2_ sym.), 1541 (NNO_2_ asym.); ^1^H NMR, δ (DMSO-*d*_6_): 3.74 (dd, *J* = 10.1, 4.2, 4H, 2 × OCH_2_CH_2_Cl), 3.81 (dd, *J* = 6.1, 3.9, 4H, 2 × OCH_2_CH_2_Cl), 4.12 (s, 4H, NCH_2_CH_2_N), 5.24 (s, 4H, 2 × OCH_2_NNO_2_). ^13^C NMR δ (DMSO-*d*_6_): 43.5 (2 × CH_2_Cl), 47.8 (2 × CH_2_NNO_2_), 69.5 (2 × OCH_2_), 79.6 (2 × OCH_2_N). Found %: C 28.74; H 4.94; N 16.65. C_8_H_16_Cl_2_N_4_O_6_. Calculated %: C 28.67; H 4.81; N 16.72.

**2-Nitraza-4-oxa-5-(azidomethyl)-6-azidohexane (3a)** (**General Procedure**). A mixture of compound **2a** (11.0 g, 50.7 mmol), NaN_3_ (16.5 g, 253.2 mmol), tetrabutylammonium bromide (TBAB, 4.08 g, 36.5 mmol) and water (150 mL) was heated under reflux overnight. After the mixture was cooled to room temperature, reaction mixture was extracted with benzene (2 × 70 mL.) The organic layer was washed with H_2_O (2 × 30 mL) and dried with MgSO_4_. Solvent was removed in vacuo to give the product **3a** in 87% yield (10.15 g) as a light-yellow mobile liquid: *d*_4_^19^ = 1.3130 g cm^−3^; *n*_D_^22^ = 1.5096. IR, ν, cm^−1^: 1087 (C-O-C), 1299 (NNO_2_ sym.), 1533 (NNO_2_ asym.), 2103 (N_3_); ^1^H NMR, δ (DMSO-*d*_6_): 3.36–3.52 (m, 7H, CH_3_NNO_2_, 2 × CH_2_N_3_), 3.94 (m, 1H, OCH(CH_2_N_3_)_2_), 5.35 (s, 2H, O_2_NNCH_2_O). ^13^C NMR δ (DMSO-*d*_6_): 37.7 (CH_3_NNO_2_), 51.4 (2 × CH_2_N_3_), 77.2 (OCH), 79.4 (NCH_2_O). ^14^N NMR δ (DMSO-*d*_6_): −27.7, −132.8, −175.4. Found %: C 26.61; H 4.36; N 48.30. C_5_H_10_N_8_O_3_. Calculated %: C 26.09; H 4.38; N 48.68.

**2-Nitraza-4-oxa-6,7-diazidoheptane (3b).** Following the same procedure, 2-nitraza-4-oxa-6,7-dibromoheptane (**2b**) gave the desired product in 80.6% yield, as a light-yellow mobile liquid: *d*_4_^19^ = 1.3188 g cm^−3^, *n*_D_^22^ = 1.5129. IR, ν, cm^−1^: 1092 (C-O-C), 1297 (NNO_2_ sym.), 1529 (NNO_2_ asym.), 2103 (N_3_). ^1^H NMR, δ (DMSO-*d*_6_): 3.39 (s, 3H, CH_3_NNO_2_), 3.49 (m, 2H, CH_2_N_3_), 3.70 (ddd, *J* = 17.4, 10.4, 5.5, 2H, CH_2_O), 3.92 (ddd, *J* = 11.2, 7.2, 4.1, 1H, CHN_3_), 5.24 (s, 2H, O_2_NNCH_2_O). ^13^C NMR δ (DMSO-*d*_6_): 37.7 (CH_3_NNO_2_), 50.7 (CH_2_N_3_), 60.1 (CHN_3_), 69.0 (OCH_2_), 80.3 (NCH_2_O). ^14^N NMR δ (DMSO-*d*_6_): −27.5 −134.4, −171. Found %: C 26.59; H 4.10; N 48.59. C_5_H_10_N_8_O_3_. Calculated %: C 26.09; H 4.38; N 48.68.

**2-Nitraza-4-oxa-6,6-bis(azidomethyl)-7-azidoheptane (3c).** Following the same procedure, 2-nitraza-4-oxa-6,6-bis(bromomethyl)-7-bromoheptane (**2c**) gave the desired product in 93.2% yield, as a light-yellow liquid: *d*_4_^19^ = 1.3117 g cm^−3^, *n*_D_^22^ = 1.5262. IR, ν, cm^−1^: 1092 (C-O-C), 1298 (NNO_2_ sym.), 1534 (NNO_2_ asym.), 2103 (N_3_). ^1^H NMR, δ (DMSO-*d*_6_): 3.38 (s, 9H, CH_3_NNO_2_, 3 × CH_2_N_3_), 3.44 (s, 2H, OCH_2_C), 5.21 (s, 2H, O_2_NNCH_2_O). ^13^C NMR δ (DMSO-*d*_6_): 37.8 (CH_3_NNO_2_), 44.0 (^t^C), 51.4 (3 × CH_2_N_3_), 68.0 (OCH_2_), 80.4 (NCH_2_O). ^14^N NMR δ (DMSO-*d*_6_): −27.5, −133.2, −171.5. Found %: C 28.48; H 4.01; N 51.92. C_7_H_13_N_11_O_3_. Calculated %: C 28.10; H 4.38; N 51.49.

**1,12-Diazido-3,10-dioxa-5,8-dinitrazadodecane (3d).** Following the same procedure, 1,12-dichloro-3,10-diox-5,8-dinitrazadodecane (**2d**) gave the desired product (95.2%), as a light-yellow liquid: *d*_4_^19^ = 1.3655 g cm^−3^, *n*_D_^22^ = 1.5185. IR, ν, cm^−1^: 1082 (C-O-C), 1275 (NNO_2_ sym.), 1539 (NNO_2_ asym.), 2104 (N_3_). ^1^H NMR, δ (DMSO-*d*_6_): 3.43 (m, 4H, 2 × OCH_2_CH_2_N_3_), 3.73 (m, 4H, 2 × OCH_2_CH_2_N_3_), 4.12 (s, 4H, NCH_2_CH_2_N), 5.23 (s, 4H, 2 × OCH_2_NNO_2_). ^13^C NMR δ (DMSO-*d*_6_): 47.8 (2 × CH_2_N_3_), 50.0 (2 × CH_2_NNO_2_), 68.1 (2 × OCH_2_), 79.6 (2 × OCH_2_N). ^14^N NMR δ (DMSO-*d*_6_) −30.5, −133.3. Found %: C 27.67; H 4.61; N 40.14. C_8_H_16_N_10_O_6_. Calculated %: C 27.59; H 4.63; N 40.22.

**1,3-Diazidopropan-2-ol (12).** Following the same procedure, 1,3-dichloropropan-2-ol **11** gave the desired product (99%), as a light-yellow liquid. The spectroscopic data are identical with those reported [36,37,38].

**N-(3-Hydroxy-2-(hydroxymethyl)-2-methylpropyl)-N-methylnitramine (8)**. A mixture of 3-methyl-3-(methylnitramino)methyl oxetane [35] (**7**, 10.9 g, 68 mmol), H_2_SO_4_ (3.7 mL, 70 mmol), water (10 mL) and dioxane (24 mL) was heated at 60 °C for 4 h. After the mixture was cooled to room temperature, it was neutralized with Na_2_CO_3_, and extracted with boiling chloroform (2 × 50 mL), and the combined organic layers were dried (MgSO_4_) and evaporated to a slightly yellow oil, which crystallized on standing; yield: 10.3 g (57.8%); mp 82–83 °C (CHCl_3_). IR, ν, cm^−1^: 1051 (2 × CH_2_OH), 1280 (NNO_2_ sym.), 1503 (NNO_2_ asym.); ^1^H NMR (DMSO-*d*_6_), δ: 0.78 (s, 3H, CH_3_C), 3.26 (m, 4H, 2 × CH_2_OH), 3.38 (s, 3H, CH_3_NNO_2_), 3.78 (s, 2H, CH_2_NNO_2_), 4.62 (t, *J* = 5.0, 2H, 2 × OH). ^13^C NMR (DMSO-*d*_6_), δ: 17.34 (CH_3_C), 40.6 (NCH_3_), 43.1 (^t^C), 55.6 (CH_2_N(NO_2_)), 64.5 (CH_2_OH). Found %: C 40.53; H 8.11; N 15.97. C_6_H_14_N_2_O_4_. Calculated %: C 40.44; H 7.92; N 15.72.

**2-Methyl-2-(methyl(nitramino)methyl)propane-1,3-diyl bistosylate (9).** Compound **8** (1.6 g, 8.53 mmol) in pyridine (10 mL) was cooled in an ice–salt bath. To this solution, TsCl (4.78 g, 25 mmol) was added at 0 °C. After addition, the reaction mixture was stirred for 30 min at 0 °C, allowed to warm to room temperature with stirring and then left for 3 days. The mixture was diluted with conc. HCl (25 mL) in an ice–salt bath. The resulting aqueous solution was removed from the sticky solid precipitate, which was dissolved in CH_2_Cl_2_ (20 mL). The organic solution was washed with water (2 × 10 mL), saturated Na_2_CO_3_ (2 × 5 mL) and water (3 × 5 mL), dried over Na_2_SO_4_, filtered, concentrated by rotary evaporation, and purified by recrystallization (EtOH) to afford colorless solid **9** (3.58 g, 86.2%): mp 96–97 °C. IR, ν, cm^−1^: 1179 (-O-SO_2_-), 1361 (-O-SO_2_-), 1280 (NNO_2_ sym.), 1512 (NNO_2_ asym.). ^1^H NMR, (DMSO-*d*_6_), δ: 0.85 (s, 3H, CH_3_C), 2.44 (s, 6H, CH_3_Ar), 3.23 (s, 3H, CH_3_NNO_2_), 3.77 (s, 2H, CH_2_NNO_2_), 3.88 (q, *J* = 9.8, 4H, 2CH_2_OTs), 7.49 (d, *J* = 8.2, 4H, Ar), 7.74 (d, *J* = 8.3, 4H, Ar). ^13^C NMR (DMSO-*d*_6_), δ: 16.5 (CH_3_C), 21.1 (CH_3_Ar), 40.9 (CH_3_NNO_2_), 41.2 (^t^C), 55.6 (CH_2_NNO_2_), 71.3 (CH_2_OTs), 127.6 (Ar), 130.2 (Ar), 131.7 (Ar), 145.2 (Ar). Found %: C 49.41; H 5.40; N 5.66. C_20_H_26_N_2_O_8_S_2_. Calculated %: C 49.37; H 5.39; N 5.76.

**N-(3-Azido-2-(azidomethyl)-2-methylpropyl)methylnitramine (10).** To a solution of sodium azide (11.05 g, 170 mmol) in DMSO (120 mL) was added compound **9** (27 g, 55.5 mmol). The resulting solution was then left to stir at 110 °C for 8 h. The reaction was then cooled to room temperature, and water (250 mL) was added slowly. The resulting emulsion was extracted with a mixture of ethyl acetate/petroleum ether (1:2, 4 × 80 mL). The organic layers were combined, washed with water (8 × 10 mL) and dried (MgSO_4_), and the solvent removed in vacuo to afford a flaxen oil; yield: 12.53 g (54.9%); *d*_4_^19^ = 1.2622 g cm^−3^; n_D_^22^ = 1.5268. IR, ν, cm^−1^: 1279 (NNO_2_ sym.), 1514 (NNO_2_ asym.), 2104 (N_3_); ^1^H NMR (DMSO-*d*_6_) δ: 0.94 (s, 3H, CH_3_C), 3.36 (s, 3H, CH_3_NNO_2_), 3.42 (s, 4H, 2 × CH_2_N_3_), 3.77 (s, 2H, CH_2_NNO_2_). ^13^C NMR (DMSO-*d*_6_), δ: 19.2 (CH_3_C), 41.8 (CH_3_NNO_2_), 42.3 (^t^C), 56.0 (CH_2_N_3_), 57.3 (CH_2_NNO_2_). ^14^N NMR (DMSO-*d*_6_), δ: −26.9, −134.2, −174.8. Found %: C 31.99; H 5.08; N 47.93. C_6_H_12_N_8_O_2_. Calculated %: C 31.58; H 5.30; N 49.10.

**1,3-Diazido-2-nitroxypropane (13).** At −10 °C, a solution of compound **12** (8.78 g, 61.8 mmol) in CH_2_Cl_2_ (12 mL) was added dropwise slowly to a mixture of nitric acid (98%, 11 mL), acetic anhydride (12 mL) and CH_2_Cl_2_ (10 mL). The solution was stirred at −10 °C for 20 min then poured into ice water (80 mL). Organic layer was separated, and water was extracted using CH_2_Cl_2_ (3 × 6 mL). The combined organic layers were then washed with water (2 × 10 mL), aqueous sodium bicarbonate solution (3 × 10 mL), and water (2 × 10 mL) again. The organic solution was dried (MgSO_4_), filtered and evaporated to a colorless oil; yield: 10.5 g (90.7%); *n*_D_^22^ = 1.4985. ^1^H NMR, δ (*J*, 300.13 MHz, DMSO-*d*_6_): 3.74 (ddd, *J* = 20.3, 13.8, 5.1, 4H, 2 × CH_2_N_3_), 5.36–5.49 (m, 1H, CHONO_2_). Found %: C 19.40; H 2.72; N 52.35. C_3_H_5_N_7_O_3_. Calculated %: C 19.26; H 2.69; N 52.40.

## 4. Conclusions

We have described a novel protocol leading to functionalized dialkyl ethers bearing both azido and nitramino groups. The methodology described is significant as (i) it is easy to implement and gives targeted products with good to excellent yields; (ii) it uses commercially available or easy-to-prepare precursors; (iii) the process is scaled to produce at least tens of grams of products.

All azido plasticizers of this study are more thermally stable, less volatile and have a more acceptable sensitivity to impact than benchmark nitroglycerin. Most of these azides have low glass transition temperatures, which makes them attractive as possible plasticizers of frost-resistant energetic materials. All the studied azides are capable of self-sustained combustion.

For the selected azides, it is shown that they can be used as plasticizers of **NC** and polyethere uretane polymer. With a plasticizer content of up to 70%, the plasticizer–polymer compositions are in a highly elastic state in a wide temperature range.

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
