# Peer review of "Promising Oxygen- and Nitrogen-Rich Azidonitramino Ether Plasticizers for Energetic Materials"

_molecules, 2022, doi:10.3390/molecules27227749_

Round 1

Reviewer 1 Report

The present manuscript developed a simple, mild and general method for the preparation of alkyl nitramines bearing a halogenoalkoxylic moiety, which were also further applied in their synthesis of azidoalkoxyl alkyl nitramines as potential energetic plasticizers. Judging from the data reported, the potential properties of some potential energetic plasticizers obtained in this work are good (based on some key factors such as acceptable thermal stability, less volatile and low glass transition temperatures). Considering the simplicity of the synthesis method, the future application of this work is promising. This manuscript itself is rich in research content, containing both synthetic studies and thermal property discussions. Moreover, different synthetic methods towards the diazide or triazide compounds have been developed, starting from various starting materials with various leaving groups, which have important reference significance for future synthesis and design of similar compounds. Therefore, I personally believe it has already met the publishing standard of Molecules and I suggest accepting this manuscript after a minor revision based on two issues listed below is finished:

1. The authors should fully compare their obtained structures (potential energetic plasticizers) with the traditional/reported structures. A complete table with more detailed information listed is suggested. Or in the separated tables, more currently applied/reported energetic plasticizers are suggested to be compared with the newly reported ones in this manuscript. This will be very helpful for a clearer conclusion and readers will be able to get complete and detailed information in this research area.

2. Please note the revision of some details, such as the literature problem shown in line 339. The quality of Figure 2 looks rough. It is recommended to use relevant software to improve the drawing.

Author Response

Dear Editor,

We thank the referees for their careful consideration of our manuscript, for the expressed wishes and for the critical comments. All necessary changes have been made in the text (the changes that I have made during the revision are highlighted with a yellow background). A point by point list of comments is given below:

Reviewer A

Comment 1: The authors should fully compare their obtained structures (potential energetic plasticizers) with the traditional/reported structures. A complete table with more detailed information listed is suggested. Or in the separated tables, more currently applied/reported energetic plasticizers are suggested to be compared with the newly reported ones in this manuscript. This will be very helpful for a clearer conclusion and readers will be able to get complete and detailed information in this research area.

Response: An additional section has been added to the manuscript, including Table 8, where some properties of the compounds of our study are listed, and a short comment to it. The new table is an addition to the tables previously placed in the manuscript.

Comment 2: Please note the revision of some details, such as the literature problem shown in line 339. The quality of Figure 2 looks rough. It is recommended to use relevant software to improve the drawing.

Response: In the paragraph on lines 332-340 there is only one reference, namely [53]. All other superscript digits are not references, but there is power for the corresponding digit, for example, 1015.07 is 10 to the power of 15.07.

Reviewer 2 Report

The authors present on the synthesis and analysis of novel plasticizers for use in energetic formulations. Ultimately the authors showed that a variety of novel plasticizers can be achieved via a simple and scalable method that was developed over the course of their study. There work is very thorough and complete with the attached SI. This is a very nice paper and should be published with very minor edits, see below.

Page 6 line 210 – alter text to say “differential scanning calorimetric (DSC)” since this is the first time that you mention DSC

Page 6 line 228 – “It is well known that some compounds crystallize poorly.” This is an incomplete sentence and should be changed to include something like “It is well known that some plasticizer compounds crystallize poorly; for example, the melting point of nitroglycerin is +13°C.”. The sentences from line 228 to 232 just need to be reworked to flow a little better.

Page 7 line 256 – Change “thermogravimetry” to “thermogravimetric analysis (TGA)”

Page 9 line 301 – You say that the compounds evaporated completely (…despite the fact that the experiment was carried out in aluminum crucibles closed with a lid, all samples almost completely evaporated to the decomposition temperature…); were the crucibles used hermetically sealed? If the samples were not hermetically sealed, then it is not surprising that there was mass loss observed over the course of the analysis. This was also not mentioned in the Materials and Methods section, besides where the authors talk about the DSC set up. You should clarify if the experiment was different or edit to text.

Page 10 line 339 – Fix reference (Error! Reference source not found.)

Page 13 line 453-454 – Combine these paragraphs.

Page 15 line 496-497 – Combine these paragraphs.

Page 15 line 498 – Change the text from “The percentage of such plasticizers can reach 70%. At such concentrations of azides, the binder is in a highly elastic state over a wide temperature range.” to “The percentage of such plasticizers can reach 70%, at which point the binder is in a highly elastic state over a wide temperature range.

Page 16 line 531 – Should be “cup” not “cap”

Please include your safety disclaimer “Caution! Although we have encountered no difficulties during preparation and handling of these compounds, they are potentially explosive energetic materials. Manipulations must be carried out by using appropriate standard safety precautions.” in the SI as well.

Author Response

Reviewer B

Comment 1: Page 6 line 210 – alter text to say “differential scanning calorimetric (DSC)” since this is the first time that you mention DSC.

Response: This is corrected.

Comment 2: Page 6 line 228 – “It is well known that some compounds crystallize poorly.” This is an incomplete sentence and should be changed to include something like “It is well known that some plasticizer compounds crystallize poorly; for example, the melting point of nitroglycerin is +13°C.”. The sentences from line 228 to 232 just need to be reworked to flow a little better.

Response: The specified phrase, as well as related ones, have been corrected. Now there is the following text: It is well known that some compounds are not prone to crystallization and remain in a supercooled state for a long time. This may be due to crystallization conditions, or to the ability of the compound to form polymorphs that give eutectic mixtures. For example, during spontaneous crystallization of NG, a stable phase with a melting point of +13 °C is formed [43].

Comment 3. Page 7 line 256 – Change “thermogravimetry” to “thermogravimetric analysis (TGA)”

Response: This is corrected.

Comment 4: Page 9 line 301 – You say that the compounds evaporated completely (…despite the fact that the experiment was carried out in aluminum crucibles closed with a lid, all samples almost completely evaporated to the decomposition temperature…); were the crucibles used hermetically sealed? If the samples were not hermetically sealed, then it is not surprising that there was mass loss observed over the course of the analysis. This was also not mentioned in the Materials and Methods section, besides where the authors talk about the DSC set up. You should clarify if the experiment was different or edit to text.

Response: Unfortunately, heating of this type of compounds in hermetically sealed crucibles is not acceptable, since in such conditions the autocatalytic process leads to a thermal explosion.

Comment 5: Page 10 line 339 – Fix reference (Error! Reference source not found.)

Response: In the paragraph on lines 332-340 there is only one reference, namely [53]. All other superscript digits are not references, but there is power for the corresponding digit, for example, 1015.07 is 10 to the power of 15.07.

Comment 6: Page 13 line 453-454 – Combine these paragraphs.

Response:  This is corrected.

Comment 7: Page 15 line 496-497 – Combine these paragraphs.

Response: This is corrected.

Comment 8: Page 15 line 498 – Change the text from “The percentage of such plasticizers can reach 70%. At such concentrations of azides, the binder is in a highly elastic state over a wide temperature range.” to “The percentage of such plasticizers can reach 70%, at which point the binder is in a highly elastic state over a wide temperature range.

Response: This is corrected.

Comment 9: Page 16 line 531 – Should be “cup” not “cap”

Response: This typo has been corrected.

Comment 10: Please include your safety disclaimer “Caution! Although we have encountered no difficulties during preparation and handling of these compounds, they are potentially explosive energetic materials. Manipulations must be carried out by using appropriate standard safety precautions.” in the SI as well.

Response: This phrase is duplicated in the SI. This correction is not highlighted with a yellow marker, as it is usually not required when editing SI.

Reviewer 3 Report

The current work described a novel protocol leading to functionalized dialkyl ethers bearing both azido and nitramino groups. An elaborated analysis was performed including impact sensetivity,  phase transition, volatility, thermal analysis.

Several questions should be addressed before publication.

1. Some references are incorrectly quoted, such as, Page 10.

2. How do the authors think the effects of energetic polymers on its ignition safety of composite materials?

3. How these energetic polymers affect the inteface bonding properties with energetic crystals?

4. Several sentences are hard to understand, like, Line 58 'on the other hand...'

Author Response

Reviewer C

Comment 1: Some references are incorrectly quoted, such as, Page 10.

Response: In the paragraph on lines 332-340 there is only one reference, namely [53]. All other superscript digits are not references, but there is power for the corresponding digit, for example, 1015.07 is 10 to the power of 15.07.

Comment 2: How do the authors think the effects of energetic polymers on its ignition safety of composite materials?

Response: The ignition safety of composite materials is a cumulative characteristic that depends on all its components. The required indicators are usually easily achieved by the introduction of special additives. Clarification of these issues needs special research.

Comment 3: How these energetic polymers affect the inteface bonding properties with energetic crystals?

Response: Both energetic polymers and energetic crystals contain polar groups that contribute to the formation of intermolecular interactions. This ensures the formation of a more homogeneous and stable composite materials. However, each specific case needs to be studied.

Comment 4: Several sentences are hard to understand, like, Line 58 'on the other hand...'

Response: I regret that there is no indication of specific phrases in our manuscript, without which it is impossible to make corrections. “on the other hand” is a common idiomatic phrase.

Reviewer 4 Report

General comments:

In this manuscript, authors report oxygen- and nitrogen-rich azidonitramino ether plasticizers for energetic materials. It is not publishable in its current form.

Comments on MS.

1.       In the abstract it is mentioned that “All azidonitramino ethers are liquid with excellent energetic performance” however detonation properties (detonation velocity and detonation pressure) or other important physiochemical properties (density, enthalpy of formation) are not reported in the MS.

2.       Abstract: “A simple, mild and general method has been developed” In fact, these are some of the common transformations in the field of energetic materials. These compounds are easily accessible.

3.       Recent literature is not cited in the manuscript and a comparison is not made with the well-established compounds. Recent reports on plasticizers: J. Org. Chem. 2021, 86, 6371−6380.

4.       Experimental section (elemental analysis): For compound 3b, N numbers are missing. For compound 2b, carbon is 1% higher (low purity).  No elemental analysis or HRMS was reported for 2a, 2d, 3d, and 13.

5.       Decomposition results (DSC-TGA measurements) are measured at 10°C/min. At the heating rate of 5°C/min, decomposition temperatures for compounds will be lower.

6.       Impact sensitivity (IS) testing: IS data do not give any idea about the sensitivity of the compounds. How hard you must hit it with the hammer (force in J) to make them explode? Sensitivity to friction is not discussed.

7.       Heat (or enthalpy) of formation and detonation parameters should be reported. 

Author Response

Comment 1: In the abstract it is mentioned that “All azidonitramino ethers are liquid with excellent energetic performance” however detonation properties (detonation velocity and detonation pressure) or other important physiochemical properties (density, enthalpy of formation) are not reported in the MS.

Response: The Table 8 with important properties for energetic plasticizers has been added to the manuscript. Detonation velocity and pressure are important for explosives, not plasticizers. What is really important for the plasticizer is that it plasticizes the necessary polymers. This property is demonstrated in our manuscript.

Comment 2: Abstract: “A simple, mild and general method has been developed” In fact, these are some of the common transformations in the field of energetic materials. These compounds are easily accessible.

Response: Indeed, a narrow range of known reactions has been used for many years to produce energetic compounds. However, when synthesizing target compounds with new combinations of substituents, these known reactions need to be adapted to these new objects. This was done in our work. It should be noted that the simpler the experimental methodology, the more likely the target product is to start being produced by industry.

Comment 3: Recent literature is not cited in the manuscript and a comparison is not made with the well-established compounds. Recent reports on plasticizers: J. Org. Chem. 2021, 86, 6371−6380.

Response: A literary search shows that in recent years, an average of ten papers concerning energetic plasticizers have been published annually. Unfortunately, it is not possible to quote all the articles. Each author is forced to cite only a limited number of references, using those that he applies in the process of work. Since our study concerns aliphatic nitramines with azide groups, only close references (but not all !) have been used.

Comment 4: Experimental section (elemental analysis): For compound 3b, N numbers are missing. For compound 2b, carbon is 1% higher (low purity).  No elemental analysis or HRMS was reported for 2a, 2d, 3d, and 13.

Response: Our attempts to make HRMS for these compounds were unsuccessful. Even the definition of elemental analysis sometimes causes problems. It is not clear what kind of error you noticed in the number N for compound 3b. Despite some overestimation of carbon in the analysis, the NMR spectra of compound 2b are of good quality. I'm sorry, but elementary analysis for 2a, 2d, 3d, and 13 forgot to enter. Now this misunderstanding has been eliminated.

Comment 5: Decomposition results (DSC-TGA measurements) are measured at 10°C/min. At the heating rate of 5°C/min, decomposition temperatures for compounds will be lower.

Response: There is no doubt that decomposition temperature depends on the heating rate. Typically, DSC measurements are used to obtain primary information about the decomposition temperature at a heating rate of 10°C/min or 5°C/min. However, the value obtained in this way serves only for relative comparison with a benchmark compound; we used NG. More informative is the definition of the decomposition rate constants, which is carried out at different heating rates (see Figure 4 and the text to it). We have conducted both a primary and a thorough study of the decomposition of compounds in this work.

Comment 6: Impact sensitivity (IS) testing: IS data do not give any idea about the sensitivity of the compounds. How hard you must hit it with the hammer (force in J) to make them explode? Sensitivity to friction is not discussed.

Response: The impact sensitivity depends on the testing method. There are several generally accepted methods for determining impact sensitivity. Some research laboratories prefer one method, others prefer another. It should be noted that even when using the same method, different researchers may get different results. To obtain objective information, the data for the test compound is always compared with a benchmark compound; we used NG. The result, whether it is represented in J or in %, is relative. The results of determining sensitivity to friction for liquid compounds are even more difficult to reproduce, and the values obtained are incorrect. We consider it unethical to produce and publish incorrect data.

Comment 7: Heat (or enthalpy) of formation and detonation parameters should be reported.

Response: The enthalpy of formation and some other characteristics are added to Table 8. As I mentioned above, detonation parameters are not significant for energetic plasticizers.

Round 2

Reviewer 4 Report

Attached as a pdf file. 

Author Response

Comment 1: Since authors had difficulty finding the errors. I have attached the screen shots in the comments. Please check.

Response: Thank you for clarifying the situation. The specified oversight has been eliminated.

Comment 2: 13C NMR of compound 2b has extra carbon peaks (shown in arrows). Extra carbon content is reflected in the EA. Therefore, I will say it has not very high purity.

Response: We made a fresh sample and re-made NMR spectra and determined elemental analysis. We have made corrections in accordance with the new data, copies of the spectra have been replaced.

Comment 3: I agree that it is only possible to include limited number of references. However, more references can be included if authors reduce the number of self-citations. Reviewer 1 also suggested to compare the properties with the newly reported ones in this manuscript.

Response: I believe that the references used should be, first of all, as close as possible to the issue under discussion and help the reader, if necessary, to get additional information about the methodology used. It does not matter who wrote the references used, by third-party authors or the authors of the same papers. The main thing is to create conditions for quickly obtaining information on a specific topic. Guided by this, our list of references has been compiled. At the same time, I note that we are far from violating the generally accepted limit of self-citation. I also note that many papers are published in which the authors suggest that the compounds synthesized by them can be used as plasticizers, but no data confirming this is provided. In particular, the reference previously proposed to us for consideration (J. Org. Chem. 2021, 86, 6371-6380) unfortunately does not contain specific information about the properties of the compounds described in it, confirming their ability to plasticize any polymers. Reviewer 1 suggested " The authors should fully compare their obtained structures (potential energetic plasticizers) with the traditional/reported structures", which we did earlier in Table 8.

Comment 4: Authors have not provided the details of the heat of formation calculations. How the values are obtained.

Response: To estimate the enthalpy of formation, we use an additive scheme. This time-tested approach gives good results, while being easy to implement. The corresponding indication and links are added to the text.

Comment 5: Line 605, mp 42÷43 °C (Et2O). Change “÷” to “-“.

Response: This is corrected.

Comment 6: Line 577, (8.6 g 66.67 mmol). Add comma after “g”.

Response: This is corrected.

Comment 7: 1HNMR,δ, (DMSO-d6) Add “:” after the bracket end.

Response: This is corrected.

Comment 8: Line 662. Define TosCl,

Response: We made a typo. Must be TsCl. The definition for TsCl is given in line 176.

Comment 9: Table 2. Change comma to point dot in change in heat capacity values.

Таблица 2. Измените запятую на точку при изменении значений теплоемкости.

Response: This is corrected.